# Resveratrol and Resveratrol-Aspirin Hybrid Compounds as Potent Intestinal Anti-Inflammatory and Anti-Tumor Drugs

**DOI:** 10.3390/molecules25173849

**Published:** 2020-08-24

**Authors:** Mohamed Salla, Vrajesh Pandya, Khushwant S. Bhullar, Evan Kerek, Yoke Fuan Wong, Robyn Losch, Joe Ou, Fahad S. Aldawsari, Carlos Velazquez-Martinez, Aducio Thiesen, Jason R. B. Dyck, Basil P. Hubbard, Shairaz Baksh

**Affiliations:** 1Department of Biochemistry, Faculty of Medicine and Dentistry, University of Alberta, 113 Street 87 Avenue, Edmonton, AB T6G 2E1, Canada; salla@ualberta.ca (M.S.); vrajeshk@ualberta.ca (V.P.); losch@ualberta.ca (R.L.); zo2@ualberta.ca (J.O.); 2Department of Pharmacology, Faculty of Medicine and Dentistry, University of Alberta, 113 Street 87 Avenue, Edmonton, AB T6G 2E1, Canada; bhullar@ualberta.ca (K.S.B.); kerek@ualberta.ca (E.K.); jason.dyck@ualberta.ca (J.R.B.D.); bphubbar@ualberta.ca (B.P.H.); 3Department of Pediatrics, Faculty of Medicine and Dentistry, University of Alberta, 113 Street 87 Avenue, Edmonton, AB T6G 2E1, Canada; yokefuan@ualberta.ca; 4Faculty of Pharmacy and Pharmaceutical Sciences, University of Alberta, 113 Street 87 Avenue, Edmonton, AB T6G 2E1, Canada or fahad386@gmail.com (F.S.A.); velazque@ualberta.ca (C.V.-M.); 5Saudi Food and Drug Authority Laboratories, 3292 Northern Ring Road, Riyadh 13312, Saudi Arabia; 6Department of Laboratory Medicine and Pathology, Faculty of Medicine and Dentistry, University of Alberta, 113 Street 87 Avenue, Edmonton, AB T6G 2E1, Canada; athiesen@ualberta.ca; 7Departments of Oncology, Faculty of Medicine and Dentistry, University of Alberta, 113 Street 87 Avenue, Edmonton, AB T6G 2E1, Canada; 8Member, Cancer Research Institute of Northern Alberta and Women and Children’s Health Research Institute, Edmonton, AB T6G 2E1, Canada; 9BioImmuno Designs, Inc., 4560 TEC Centre, 10230 Jasper Avenue, Edmonton, AB T5J 4P6, Canada

**Keywords:** resveratrol, cancer, inflammation, NFκB, AMPK, sirtuin, inflammatory bowel disease

## Abstract

Resveratrol (3,4,5-Trihydroxy-trans-stilbene) is a naturally occurring polyphenol that exhibits beneficial pleiotropic health effects. It is one of the most promising natural molecules in the prevention and treatment of chronic diseases and autoimmune disorders. One of the key limitations in the clinical use of resveratrol is its extensive metabolic processing to its glucuronides and sulfates. It has been estimated that around 75% of this polyphenol is excreted via feces and urine. To possibly alleviate the extensive metabolic processing and improve bioavailability, we have added segments of acetylsalicylic acid to resveratrol in an attempt to maintain the functional properties of both. We initially characterized resveratrol-aspirin derivatives as products that can inhibit cytochrome P450 Family 1 Subfamily A Member 1 (CYP1A1) activity, DNA methyltransferase (DNMT) activity, and cyclooxygenase (COX) activity. In this study, we provide a detailed analysis of how resveratrol and its aspirin derivatives can inhibit nuclear factor kappa B (NFκB) activation, cytokine production, the growth rate of cancer cells, and in vivo alleviate intestinal inflammation and tumor growth. We identified resveratrol derivatives C3 and C11 as closely preserving resveratrol bioactivities of growth inhibition of cancer cells, inhibition of NFκB activation, activation of sirtuin, and 5’ adenosine monophosphate-activated protein kinase (AMPK) activity. We speculate that the aspirin derivatives of resveratrol would be more metabolically stable, resulting in increased efficacy for treating immune disorders and as an anti-cancer agent.

## 1. Introduction

Resveratrol (3,4′,5-trans-trihydroxystilbene), like other plant bioactive products, is a part of the defensive mechanisms that have evolved in plants to deter and protect against pathogens and cellular damage. Extensive studies have aimed at investigating the efficacy of polyphenols as anti-inflammatory and anti-tumor/chemoprevention agent [1,2]. For instance, terpenoids, which is one of the four major classes of secondary plant metabolites, reported to exert selective anti-proliferative effects in colon cancer cells with no toxicity on normal human intestinal cells [3,4,5]. Resveratrol is a naturally occurring phytoalexin present in grapes, peanuts, and other plants [6]. Having a simple structure and low molecular weight, resveratrol can target key proteins such as cyclooxygenase (COX) enzymes, AMPK [1,7], and NFκB, factors strongly associated with inflammation and carcinogenesis [8]. In addition, resveratrol can also regulate other targets such as Phosphoinositide 3-kinases (PI3)-kinases, Sirt1, DNA methyltransferases [9], and others. Resveratrol is gaining interest in combinational therapy for many diseases, including cancer, as it targets a plethora of pathways, leading to the “abnormal” state. It is relatively non-toxic as compared to other chemotherapeutic drugs. However, despite its interesting biological properties, resveratrol is required in high doses to induce apoptosis of cancer cells or alter metabolism and its biological activity is limited by its photosensitivity and metabolic instability [10]. Resveratrol is also limited by its low bioavailability owing to its extensive metabolism in the liver, with a serum half-life of about 14 min [11]. Extensive research has been done to decrease these limitations by producing resveratrol derivatives that might have more potent biological activities at lower doses and with higher bioavailability.

Non-steroidal anti-inflammatory drugs (NSAIDs) have recently gained wide scientific interest for their promising chemo-preventative properties. Aspirin, one of the most common NSAIDs, is of particular interest in this regard. It is currently recommended for cancer prevention in patients with a low risk of bleeding [12]. Interestingly, evidence for its chemo-preventative role comes from studies of patients with prior history of colorectal cancer (CRC) and colorectal adenomas [13,14] in which aspirin reduced the number of newly formed adenomas as compared to control patients that were aspirin free. Other NSAIDs such as rofecoxib showed effective cancer prevention properties but toxicity associated with the daily doses led to a drawback in their use [15]. Furthermore, celecoxib, a COX-2 selective inhibitor of the NSAID family, also inhibited adenoma formation in patients with familial adenomatous polyposis (FAP) [16]. Aspirin appears to be effective mainly due to (but not limited to) modulation of prostaglandin biology. Aspirin can also be acetylated to influence several other molecules such as tumor necrosis factor alpha (TNF-α), resolvins [17], PI3K [18], NFκB [19], caspase-3 [20] and 6-phosphofructo-1-kinase [21]. Inflammation is critical to the rise of pre-tumor lesions and the maintenance of a tumor-favoring microenvironment. Diseases such as inflammatory bowel disease (IBD) are characterized by a state of chronic inflammation that can predispose one to a higher risk of cancer. Therefore, targeting the inflammatory component of these diseases using NSAIDs and NSAID-like molecules is of great preventative value to reduce cancer incidence. The precise mechanism(s) of NSAID action remains to be investigated with respect to stage of intervention during the carcinogenesis process and the molecular targets affected.

As mentioned earlier, the use of aspirin as a prophylactic drug remains spurious due to its adverse effects on bleeding. Salicylic acid is known to bind to serum albumin [22], and most of our derivatives are attached to a salicylate pharmacophore [23], which should increase binding to serum albumin and increase their solubility and bioavailability. With that in mind, we aimed in this study to explore the effectiveness of resveratrol-aspirin hybrid derivatives as possible therapeutic drugs against inflammation and cancer. Our results demonstrate the varied effects of adding salicylic acid at numerous positions on the resveratrol chemical structure and identified two potential viable choices for further in vivo analysis.

## 2. Results

### 2.1. Resveratrol Derivative C11 Exhibits a Greater Anti-Proliferative Potential Than Resveratrol

Resveratrol salicylate hybrids were designed by adding a carboxylic acid group adjacent to one of the phenols in the resveratrol structure [23]. The chemical structures of the more relevant derivatives for this study are shown here (Figure 1A) with others shown in Appendix A. The effect of these derivatives on cellular viability was investigated in several cancer cell lines using the MTT (3-(4,5-dimethylthiazol-2-yl)-2,5-diphenyl-tetrazolium bromide) assay that measures the reduction of MTT to formazan by NAD(P)H-dependent oxidoreductase enzymes and is used as an indirect measure for the growth rate. Interestingly, Resv-C11 derivative inhibited most of these cell lines to a greater extent than the parent resveratrol compound with IC_50_ values of 35 µM, 50 µM, and 50 µM in HCT-116 (colon cancer), PANC1 (pancreatic cancer), and A549 (lung cancer), respectively, with similar results in non-cancer cells line of BHK and NIH-3T3 lines (Figure 1B–D, Appendix A). C11 treatment results in >95% inhibition of growth by promoting a robust G1 arrest to inhibit growth. These results highlight the potential of the Resv-C11 compound as an anti-proliferative drug candidate.

### 2.2. Resveratrol Derivative C3 Consistently Inhibits NFκB Activity in Colon Cancer and Normal Cell Lines

Uncontrolled chronic inflammation is highly associated with various inflammatory diseases and cancers including IBD-driven CRC. NFκB signaling is hyperactivated in both IBD and CRC, providing an important mechanistic link between the two pathologies [22]. Therefore, we next aimed to assess the efficacy of the resveratrol derivatives in inhibiting NFκB activity by utilizing a dual-luciferase reporter assay for NFκB target gene (IL-6) in two colon cancer cell lines (HCT-116 and SW-480) and a normal mouse intestinal cell line (ModeK). Colon cancer cells are very responsive to lipopolysaccharide (LPS) treatment and display robust NFκB activities. IL-6 is a pro-inflammatory cytokine that is involved in maintaining a state of chronic inflammation by promoting the accumulation of T cells resistant to apoptosis and is associated with increased production of intracellular adhesion molecule-1 (ICAM-1) [25]. Cells were stimulated with the Toll receptor (TLR) activator, lipopolysaccharide (LPS), and the effectiveness of the drugs was measured by their ability to reverse the observed LPS-induced activation of NFκB. Based on our screening, compounds C3, C11, and C12 seemed to consistently inhibit the activity of NFκB in the three cell lines (Figure 2A–C). C3 appears to more potent than C11 in inhibiting NFκB with IC_50_ values of 10 µM vs 12 µM in HCT116 cells, 8 µM vs 35 µM in SW-480 cells, and 40 µM vs 70 µM in the normal mouse ModeK cell line. Results with other cytokines are shown in Appendix A. These findings indicate an important property of the C3 compound being more selective in colon cancer cell lines than the normal cell line with approximately 4 times lower IC_50_ values. To further confirm these results, a DNA binding assay was performed on nuclear extracts from HCT-116 cells treated in vivo with resveratrol or its derivatives at a final concentration of 100 µM. The ability of NFκB to bind the promoter of its target gene, IL-6, was significantly reduced when treated with resveratrol, C3 and C11 derivatives (Figure 2D) but not with the C10 or C12 derivative.

### 2.3. Resveratrol and Derivatives Inhibit LPS-Driven Cytokine Production in Colon Cancer Cell Lines

NFκB is a pivotal player in inflammation and IBD pathology. The use of steroids in the treatment of IBD is mainly directed towards the inhibition of NFκB activation but can influence interferon gamma (IFN-γ) response pathways [26]. Uncontrolled inflammation results in unregulated cytokine production of IL-1β, IL-2, IL-6, IL-8, IL-12, and TNF-α to mention a few [27]. Therefore, we measured the levels of several cytokines known to be involved in inflammation, especially in IBD pathogenesis, in the presence of resveratrol and derivatives upon LPS stimulation. Levels of IL-6 were reduced by the resveratrol derivatives with C3 and C12 showing slightly greater reduction (Figure 3. Moreover, HCT-116 cells demonstrated elevated levels of IL-8 in response to LPS that can be significantly reduced with resveratrol, C3, C10, C12, and to some extent C11 (Figure 3). IL-8 is the first chemokine to be studied in IBD [28] and reportedly shown to be elevated in ulcerative colitis patients. Interestingly, other cytokines including IL-10 (Figure 3) and IL-23 (Figure 3) were also reversed upon resveratrol and derivative treatment. IL-23 has been reported to be highly up-regulated in Crohn’s Disease (CD) [29] and plays an important role in stimulating the pro-inflammatory response [30]. IL-12 is produced by macrophages and dendritic cells and can enhance natural killer-mediated cytotoxicity [29] and along with IL-18 triggers IFN-γ production [31].C3 derivative showed superior inhibition of IL-12, with resveratrol and C12 being effective as well (Figure 3). Furthermore, TNF-α, which is known for enhancing NO production and triggering metalloproteinases (MMPs), was significantly reduced by all derivatives used (Figure 3F). Inhibition of TNF-α can be protective against loss of epithelial integrity [32]. GM-CSF (Figure 3G) were also significantly reduced by the derivatives used. We did not observe significant changes in IL-1β, IL-4, IL-5, IL-13, IL-17A, MIP-1α, IFN-γ (data not shown). Thus, several resveratrol derivatives can reduce cytokine production by > 50% in two colorectal cancer cell lines (HCT116 and SW480) to suggest that modification of the parent resveratrol structure does not appreciably alter the ability to interfere with inflammation-driven cytokine production.

### 2.4. Resveratrol and Derivatives Arrested HCT-116 Colorectal Cancer Cell Line at Different Stages of the Cell Cycle with C3/C11 Promoting Cell Death

An important hallmark of cancer is the dysregulation of the cell cycle components including the well-known cyclins and associated cyclin-dependent kinases (Cdk). Several Cdk inhibitors have reached clinical trials, such as Flavopiridol (blocks cdk9 transcriptional activity) [33] and palbociclib that can block cdk4/6 [34,35]. Combination therapy of Cdks inhibitors with some of the established cytotoxic agents have gained interest as well [36], adding to a possible promising role of anti-tumor drugs that target the cell cycle components. We next aimed to analyze the effects of resveratrol salicylate hybrids on cell cycle phase profiling. Interestingly, only C3 was able to significantly induce a sub-G1 (6-fold) and a G2/M (2-fold) arrest in HCT-116 cells (Figure 4A,B), indicative of possible cell death pathways being activated and/or reactive oxygen species (ROS) (generation and DNA damage. However, upon pathway-specific immunoblotting for markers of cell death, it was observed that both C3 and C11 can promote active cell death with C11 revealing a robust trigger for caspase-3 and PARP cleavage (Figure 4D). Moreover, when looking at the G2/M arrest, resveratrol also actively arrested the cells at this checkpoint (Figure 4A,B). Notably, resveratrol and C3 shifted the cells into an S-phase arrest. Furthermore, we assessed the expression of two critical components of the cell cycle control mechanism, the activities of cyclin D1 and cyclin A that can associate with and influence Cdk4 and Cdk1 functions, respectively. Several reports have shown that Cdk4 hyperactivity is associated with the overexpression of cyclin D [37] and several cancers showing elevated levels of cyclin D1 expression including breast (60% of patients), CRC (40% of patients), and prostate cancers (20% of patients) [38]. Only C3 and resveratrol reduced the expression of cyclin D1 and cyclin A (Figure 4C), confirming an important role of these compounds in the restriction of cell cycle progression in HCT-116 cells and predictably in CRC.

### 2.5. Resveratrol, C3, and C11 can Alleviate DSS-Induced Acute Intestinal Inflammation Injury

Results thus clearly indicate the potency of resveratrol and its derivatives as NFκB, growth, and cytokine production inhibitors. We have previously published on the sensitivity of the Ras association domain family 1 isoform A (*Rassf1a*)*^−/−^* animal in response to DSS-induced inflammation injury in the colon–a colitis-like model in rodents [39]. In that publication, we provided evidence to support the role of RASSF1A as a negative regulator of inflammation when, if lost, will result in uncontrolled inflammation in the colon. Thus, it is a useful and sensitive model of colitis in rodents. We can alleviate the effect of DSS-induced intestinal inflammation with the use of a resveratrol diet (containing 2 mg resveratrol/g of food) to promote a > 60–70% recovery in mice (Figure 5A,B). Histological staining confirms efficient recovery from inflammation injury to reveal an intact and healthy crypt structure, length, and proliferative capacity (PCNA stained results) (Figure 5C,D and Appendix A). The results using Resv-C9 and Resv-C11 are shown in Appendix A with additional data on body weight changes and food consumption. Furthermore, colonic inflammation triggers in vivo activation of bone marrow-derived macrophage (BMDM) that was lost in the resveratrol diet-fed animals (Figure 5E). These observations would suggest that NFκB activation is primarily the cause of intestinal inflammation injury in our DSS model, and resveratrol is sufficient to alleviate disease symptoms of colitis in mice.

### 2.6. Resveratrol, C3, and C11 can Reduce Tumor Burden in Xenograft Model

We can observe the robust anti-growth properties of several resveratrol derivatives. HCT116 cells produce very aggressive tumors in a xenograft model. We can observe a robust growth after day 15 in our model (Figure 6A). Diet-fed resveratrol was very effective in reducing this aggressive growth and so were Resv-C3 and Resv-C11 (Figure 6A). Both the rate of tumor growth and size of the tumor on day 28 were reduced, further illustrating the effectiveness of resveratrol and its novel derivatives as an effective anti-tumor agent. We did not observe any protective effect for Resv-C10 or Resv-C12.

### 2.7. Resveratrol Derivatives do Not Appear to Inhibit Sirtuin Activity Unlike Resveratrol

Resveratrol has been suggested to have broad clinical applications ranging from anti-aging properties, cardioprotective features, metabolic regulator to anti-inflammatory properties [40,41,42]. Several targets have been identified [41] including the silent mating-type information regulation 2 (sirtuin/Ser2) family of proteins, primarily SIRT1. The sirtuins function as lysine deacylases to the mediate deacetylation reaction coupled to NAD hydrolysis [43]. This provides a direct link to the energy status of the cell and hence links to AMPK. It appears as though Sirt1 activation by low doses of resveratrol is the key target of resveratrol that is subsequently responsible for AMPK activation [44]. Evidence from cell culture studies suggests that overexpression of SIRT1 reduces the acetylation of LKB1, and increases the activity of LKB1 (and hence activation of AMPK), while knock-down of SIRT1 expression increases LKB1 acetylation and reduces LKB1 (and AMPK) activity [45,46]. Thus, it is important to determine the effect of resveratrol derivatives on sirtuin and AMPK activities. Analysis of sirtuin activity using the β-NAD peptide assay revealed that parent molecule can activate sirtuin activity at 10 µM whereas all the resveratrol derivatives cannot (Figure 6B). This suggests an important difference between parent resveratrol and its derivatives and negates modulation of sirtuin activity as the primary cause of alleviation of intestinal inflammation in Figure 5 and anti-tumor effect in Figure 6B.

We next investigated the activation of AMPK in HCT116 cells using either parent resveratrol and its derivatives. Parent resveratrol can activate AMPK effectively at 100 µM and to some extent, C3 and C11 can also do that with no effect of C12 on AMPK activation. Thus, salicylate derivatives of resveratrol revealing differing in vivo biochemical properties than parent resveratrol and may have promise as more specific in vivo inhibitors of inflammation, growth, and metabolism.

## 3. Discussion

### 3.1. Salicylate Derivatives of Resveratrol Reveal Promising Biological Properties

Resveratrol has emerged into a multi-purpose natural plant-derived polyphenol that regulates diverse cellular processes ranging from energy metabolism, inflammation, cardiovascular protection, aging to the circadian rhythm [1]. For some of these, the biological targets of resveratrol have been determined but for most, the targets remain elusive. Molecular targets of resveratrol are still being uncovered [24,40,41] and modulation of multiple targets may explain the beneficial effects of resveratrol when used to ameliorate disease. Despite its robust in vivo efficacy, resveratrol suffers from a rapid and extensive metabolization into its glucuronide and sulfate conjugates as well as to the corresponding reduced products. In an attempt to obtain in vivo rapid metabolization and reduction while maintaining or enhanced biological specificity, several groups have attempted to derivatize resveratrol. This includes several monoalkoxy, dialkoxy, and hydroxy analogs of resveratrol that have been observed to be more effective in activation of PKCα; [47,48] mono, di, and tri-acetoxy resveratrol(s) were more effective at in vitro inhibition of CYP3A4 [48]; other resveratrol-conjugates for inhibition of MMP-9, DNA damage, and activation of p53-p21^CIP1/WAF1^ [49,50,51] and recently by our group the effects of aspirin-resveratrol fusions [23,24].

### 3.2. Salicylate Derivatives of Resveratrol Have More Defined Targets

We have previously documented how aspirin-resveratrol derivatives more robustly inhibit CYP1A1, DNA methyltransferase activity, and COX-1 and COX-2 activity. In this study, we further demonstrate the effectiveness of aspirin-resveratrol derivatives on inhibition of NFκB activity, growth control, and sirtuin activities (Figure 1, Figure 2 and Figure 6) while improved activation of AMPK can be observed (Figure 6C). Furthermore, we can demonstrate significant alleviation of intestinal inflammation and inhibition of tumor growth with several aspirin-resveratrol derivatives (Figure 5 and Figure 6A). Previous reports have confirmed our observations including a recent publication by Sudha et al. (2020) [52] to demonstrate tumor reduction by resveratrol and publications by Wang et al. (2019) [53] and Zhang et al. (2019) [54] that demonstrated the beneficial effect of resveratrol on DSS-induced colitis but at 80,100 mg/kg resveratrol use. In addition, cell-based modulation of follicular tumorigenesis (at 300 µM) [55] highlights the importance of our findings at 50–100 µM for our resveratrol derivatives. Because of the effectiveness of several aspirin-resveratrol derivatives towards both growth and NFκB inhibition (such as C3 and C11), these aspirin-resveratrol derivatives could be potent inhibitors of inflammation-driven malignant transformation. An analysis of oral bioavailability, glucuronide and sulfate conjugation of these aspirin-resveratrol derivatives is currently being investigated, but we speculate that the aspirin-resveratrol derivatives may be more stable than parent molecule to result in increased biological activities. Although some aspirin-resveratrol derivatives reveal similar IC_50_ values for growth and NFκB inhibition, we can start to observe differences in how they can inhibit growth. Aspirin-resveratrol derivative C3 and C11 can promote apoptosis while C11 induced a profound G1/G0 growth arrest (Figure 4). Generally, C3 has a more robust effect on inhibition of NFκB while C11 addition has a 13% greater overall survival than C3 following DSS-induced colitis (Figure 5A). Strikingly, only parent resveratrol had appreciably any effect on sirtuin activity to suggest a SIRT1-independent role in growth inhibition of the aspirin-resveratrol derivatives, or protection from inflammation injury or anti-tumor effect as observed in this study. A complete summary of the bioactive properties of our resveratrol derivatives is found in Table 1.

### 3.3. Potential Clinical Use of Salicylate Derivatives of Resveratrol?

In general, preclinical studies have yielded promising results about the benefits of resveratrol for the management of a variety of diseases related to inflammation [56] as well as for cancer [57]. However, treatment and/or prevention of most diseases in animals do not always translate to human studies, so care should be taken in our interpretation of the effectiveness of resveratrol in clinical use. Indeed, a prospective cohort study demonstrated that total urinary resveratrol metabolite concentration was not associated with inflammatory markers, cancer, or cardiovascular disease, or predictive of all-cause mortality [58], thus suggesting that there is no benefit to resveratrol supplementation in humans. However, that study investigated these outcomes in older community-dwelling adults who consume resveratrol as part of their regular diet and the amount of resveratrol consumed from the dietary sources is much lower than the therapeutic doses of resveratrol used in various clinical trials. Since most of the effects of resveratrol are dose-dependent [59], the negative findings from the observational study [58] are probably not representative of the effects of resveratrol when administered as a nutraceutical. Therefore, it is important to address the challenges that face the translation of the very promising preclinical results to real-world clinical benefits of resveratrol.

### 3.4. Can Salicylate Derivatives of Resveratrol Overcome Low Bioavailability Properties?

Perhaps one of the largest challenges associated with resveratrol is its low bioavailability [60,61,62]. Although resveratrol is highly absorbed when given orally, it has very low bioavailability due to rapid metabolism to its glucuronide and sulfate conjugates [63]. Administration of approximately 25 mg resveratrol results in plasma concentrations that are between 1 and 5 ng/mL [64], and administration of even larger amounts, up to 5 g, still only yield 500 ng/mL (2 μM) [65]. This has resulted in the concept of the “resveratrol paradox”, which describes a molecule with a very low plasma concentration that has multiple biological effects [66]. To explain this, several theories have been proposed ranging from resveratrol concentrating in certain organs [67] to the evidence provided that shows the conversion of sulfated conjugates of resveratrol back to the parent compound in certain cell types [68]. Although this latter concept presents a possible explanation for the in vivo results observed for resveratrol for the treatment of disease in humans, it needs to be investigated whether the appropriate target cells also have the ability to transport the sulfated conjugates of resveratrol into the cell to be metabolized back to the parent compound [68].

Due to the poor bioavailability of resveratrol, there remains a question as to what dose should be used for clinical studies? In the animal studies reviewed herein, effective doses have ranged from as low as 0.1 mg/kg/day up to 800 mg/kg/day. Similarly, in clinical studies, doses of resveratrol range from 10 mg/kg/day to 1000 mg/kg/day (our derivatives were utilized at 68 mg/kg every 2 days but we have observed beneficial effects at 30 mg/kg every 2 days, data not shown). This issue of the dose is particularly important as several resveratrol effects are dose-dependent. Furthermore, since resveratrol may exhibit a hormetic dose-response, this further complicates a dose selection for clinical studies [69]. Another important and related question is what method of administration should be chosen. Although resveratrol is poorly soluble in aqueous solutions [70], some preclinical studies administer resveratrol in the drinking water, which may create dose variability issues. How this hurdle will be overcome in clinical trials has yet to be addressed but the poor solubility of resveratrol may be enhanced by increasing its aqueous solubility via microparticulate systems, cyclodextrin complexes, nano-carrier systems, or even vesicular systems (reviewed in Reference [62]). Whether or not this will improve the effectiveness of resveratrol in clinical studies has yet to be tested. Because of the aforementioned reasons, the bioavailability of our aspirin-resveratrol derivatives will need to be fully investigated in order to be clinically useful. Our study provides in vitro and in vivo insights into these aspirin-resveratrol derivatives but further pharmacological analysis and pharmacokinetic analysis, as well as detailed bioavailability/tissue distribution analysis, will be required before human clinical trials are attempted.

## 4. Materials and Methods

### 4.1. Cell Culture and Transfection

The ModeK normal mouse intestinal cell line (grown in Gibco DMEM high glucose supplement with 5% FBS, 1% non-essential amino acids and 7.5 mL 1 M HEPES) was generously provided by Karen Madsen (University of Alberta, Edmonton, AB, Canada). The colon cancer cell line HCT116 (obtained from Dr. Bert Vogelstein, Baltimore, MD, USA), was maintained in RPMI 10% BGS. The colon cancer cell line SW480 (obtained from Karen Madsen [University of Alberta]) was maintained in DMEM supplemented with 10% BGS. All cells were maintained in a 37 °C/5% CO_2_. Transfections were carried out using the linear 25 kDa polymer, polyethyleneimine (PEI) obtained from Polysciences, Warrington, PA, USA (Catalog #23966–2). PEI transfections were carried out by mixing PEI/DNA in a ratio of 4 µL PEI/1 µg DNA in serum-free media to achieve efficiencies of 30–50% as described elsewhere (Foley et al., 2008) [71]. HCT116 cells were utilized for our xenograft assays as tumors form within 30 days. PANC1 pancreatic cancer, A549 lung cancer, BHK, NIH 3T3 cell lines (Cancer Research UK, now The Francis Crick Institute, London, UK); 231 breast cancer cell line (Dr. Ing Swie Going, Edmonton, AB, Canada) were obtained from collaborators as indicated. Further details are available from S.B. upon request.

### 4.2. Cell Viability

Cell viability assay was determined using the MTT [3-(4,5-dimethylthiazol-2-yl)-2,5-diphenyl-tetrazolium bromide] colorimetric assay as indicated. MTT measures respiring cells and is directly correlative to the number of cells present. Normalization was done against cells not treated drug (100% viable). Briefly, cells were seeded in 96-well plates at a density of 4 × 10^4^ per well and were allowed to attach overnight. Resveratrol or its derivatives (dissolved in 100% DMSO) were added to the corresponding 96-well plate containing 100 µL serum-free media and then incubated for 24 h at 37 °C. The media was aspirated out and the wells were washed one time with 1 × phosphate-buffered saline (PBS), followed by 100 µL of fresh media and 15 µL of 12 mM MTT. The plate was then incubated for 4 h at 37 °C. Finally, the insoluble formazan crystals in the media were removed and 50 µL of 100% DMSO added for 10–15 min. The absorbance of each well was read at 540–570 nm (we read it at 570 nm). The percentage of cell viability was correlated to that of DMSO-treated wells, which were set to a value of 100% viability. Each compound was assayed at three concentrations in triplicate and the experiment was performed three times, and the results are expressed as mean ± SD.

### 4.3. Dual-Luciferase Assay

Dual-Luciferase Reporter Assay System (DLR assay system, Promega, E1910, Wisconsin, WI, USA) was used to perform dual-reporter assays on NFκB Luciferase and Renilla Luciferase (internal control). Briefly, cells were equally seeded at a density of 3 × 10^4^ in 6-well plates and allowed to attach for 24 h. Prior to transfection, cells were washed with serum-free media 3 times. Dual transfection was carried out using PEI adding 3 µg of NFκB Luciferase construct and 60 ng of Renilla Luciferase construct as an internal transfection control. After 24 h post-transfection, cells were treated with the different drugs for 24 h as well and stimulated with LPS for 4 h in serum-free media. Cells were then lysed using the passive lysis buffer provided by the kit for 30 min on ice. Twenty µL of cell lysate was transferred in 96-well plate. Luciferase assays were analyzed based on the ratio of the Firefly/Renilla to normalize cell number and transfection efficiency.

### 4.4. Cell Cycle Analysis

For cell cycle analysis, cells were seeded in 6-well plates, allowed to grow to 60% confluency and treated with DMSO or the resveratrol derivatives for 24 h. Detached cells in the supernatants in addition to the attached cells were collected. The pellets were washed with phosphate-buffered saline (PBS) (Gibco), fixed with 70% ice-cold ethanol, and stored at -20 C overnight. Next day, ethanol fixed cells were centrifuged (500× *g*) and washed twice with 1 × PBS at 4 °C. Propidium iodide staining of the cells was performed by incubating cell pellets in DNA staining solution (0.1% Triton X-100, 100 μg/mL DNAse free RNAseA, 20µg/mL propidium iodide in 1 × PBS) at 37 °C for 30 min followed by filtration through a nylon mesh (50 µM exclusion limit) to remove cell clumps. Flow cytometric analysis was performed on a BD Accuri cell analyzer (Franklin, TN, USA). Cell cycle analysis was performed using the BD Cell Quest program (version 2, Franklin, TN, USA).

### 4.5. Interleukins/Chemokine ELISA

Chemokine/Cytokine ELISA array performed by Eve Technologies, Calgary, Canada). Briefly, cells were seeded equally in 6-well plates and allowed to adhere for 24 h then treated with the different concentrations of resveratrol and derivatives for 24 h and 500 µL of supernatant was collected and sent on dry ice to Eve Technologies for analysis of cytokine production.

### 4.6. Electrophoretic Mobility Shift Assay (EMSA)

EMSA was carried out as described previously (Gordon et al. 2013) [39]. Four micrograms nuclear extracts were prepared using NE-PER (ThermoFisher Scientific, Waltham, MA, USA) and duplex DNA specific NFκB binding site probes end-labeled with [γ-32P] ATP by T4 polynucleotide kinase. The oligonucleotide for NFκB analysis was obtained from the site on the IL-6 promoter (forward primer was AAATGTGGGATTTTCCCATGA) and IL-8 promoter (forward primer was TCAGAGGGGACTTTCCGAGAGG) [39]. Two μg nuclear extracts were incubated with duplex DNA/protein complexes were for 30 min and then separated by non-denaturing gel electrophoresis, dried onto Whatman filter paper and autoradiographed. Binding buffer was 100 mM Tris-HCL pH 6.5, 500 mM KCl, 1.2 mM EDTA, 12 mM DTT, 20% glycerol, and 1 μg Salmon Sperm DNA.

### 4.7. Mouse Experiments

All animal experiments were approved and conducted according to the animal ethics board at the University of Alberta: AUP00000218 (For cancer-related experiments) and AUP00000219 (for inflammation-related experiments). All mice were on the C57BL/6 background. For our inflammation acute model, mice (males at 23–25 g) were given 7 days of 3% dextran sodium sulphate (DSS) followed by water for 7 days for recovery with a regular chow diet. The resveratrol group mice were pre-fed for 14 days with resveratrol mixed with a chow diet at a concentration of 1 mg resveratrol/g (Human equivalent dose of 5.8 mg/kg body weight). Mice were monitored for weight changes and clinical symptoms of colitis, which were assigned a disease activity index (DAI) score as described previously (Gordon et al., 2013) [39]. Generally, if the animal has 25 g to start, at the end of the 14-day treatment with severe inflammation, the body weight is generally around 14–16 g. All animals were euthanized by CO_2_ inhalation according to protocol.

### 4.8. Tissue Handling

Tissue samples collected were for molecular analysis. Samples intended for nucleic acid analyses were immediately submerged in RNA-later post excision and allowed to incubate at 4 °C for 24 h to fully permeate the tissues. At this point, samples were removed from RNA-later and stored at −80 °C until further use. RNA was isolated from colon tissue using the Qiagen All Prep DNA/RNA spin column nucleic acid extraction kit (Hilden, Germany) according to the manufacturer’s instructions. The nucleic acid concentration was determined using Nanodrop (Thermo Scientific, Waltham, MA, USA).

### 4.9. Crypt Cell Isolation

Intestinal crypt cells were isolated as outlined previously (Xiao et al., 2007) [72]. Briefly, colons were flushed with cold 1 × PBS, cut open longitudinally, and then soaked in 1 × PBS with gentle shaking for 20 min. The colons were cut into small pieces and incubated with 0.04% sodium hypochlorite for 30 min with gentle shaking, followed by incubation at room temperature for 30 min in a solution containing 1 × PBS/1 mM EGTA/1 mM EDTA with shaking. Cells were then dislodged by pipetting the tissue up and down using a 25 mL serological pipette until the solution became cloudy. The supernatant was removed (containing the crypt cells) and centrifuged at 3000 rpm for 10 min to collect the crypt cells followed by nuclear extraction.

### 4.10. Subcutaneous Injection of Tumor Cells

HCT116 colon cancer cells were trypsinized and spun down at 1000 rpm for 5 min and resuspended in a 4:1 mix of media:matrigel (BD #354234, 10 mg/mL of LDEV-free matrix). Two hundred microliters (containing ~ 2 × 10^6^ cells) of this mixture was subcutaneously injected into the right and left flanks of athymic nude mice (Taconic Laboratories #NCRNU-M, CrTac:NCr-FoxN1Nu, Rensselaer, NY, USA) in order to determine tumor-promoting potential. Mice were monitored weekly until tumors appeared and euthanized once tumors exceeded 20 mm in diameter. Resveratrol was fed to the mice in the diet and resveratrol derivatives (C3 or C11) were injected intraperitoneally at 0.3 mmol/kg body weight every 2 days until the end of the experiment.

### 4.11. Immunoblotting

For cleaved caspase 3 and PARP, cells were treated with resveratrol or resveratrol derivatives at 100 mM for 24 h prior to lysis. Protein loading buffer containing SDS was added directly to pellets to lyse the cells followed by a brief spin down at 10,000 rpm for 1 min and application to SDS-PAGE and transfer to PVDF membrane for immunoblotting. For cleaved caspase 3, a 12.5% gel was run ¾ of the way in order to observe the small cleavage products. For PARP analysis, a 10% gel was run as per normal. For AMPK analysis, cells were lysed in RIPA buffer followed by separation on SDS-PAGE and transferred to an Amersham Protran 0.45 µm nitrocellulose membrane (GE Healthcare Life Sciences, Chicago, IL, USA) for 2 h at 90V in a Bio-rad Transblot wet transfer cell (Hercules, CA, USA). Membranes were blocked in 5% skim milk in 1 × TBST for 1 h at room temperature, washed three times and primary antibody anti-pAMPKalpha (T172) rabbit Ab [Cell Signaling Technology, Danvers, MA, USA) 1:1000 in 5% milk was added for 1 h at room temperature. Secondary antibody was removed and membranes were washed followed by ECL detection (GE Amersham ECL RPN2106, Chicago, IL, USA).

### 4.12. Tissue Histology and Immunohistochemistry

Colon samples were isolated, fixed in z-Fix (Anatech Ltd., Herculaes and paraffin-embedded. All inflammation scores were obtained utilizing blinded scoring by a gastrointestinal pathologist (Dr. Aducio Thiesen) based on the infiltration of enterocytes, neutrophils, lamina propria cellularity, crypt structure, and epithelial hyperplasia (scored as 0–2 where 2 = maximal injury) (Madsen et al., 2001) [73]. Immunohistochemistry (IHC) and hematoxylin and eosin (H&E) staining were carried out using standard techniques. Formalin-fixed, paraffin-embedded sections were de-paraffinized and re-hydrated. Antigen retrieval was done by boiling in sodium citrate buffer. Endogenous peroxidase activity was quenched with 3% H_2_O_2_. Sections were blocked in 2% BSA + 2% donkey serum for one hour at room temperature and incubated in primary antibody as indicated overnight at 4 °C. Sections were incubated in 1:500 biotinylated secondary antibody for 1 h at room temperature and signal amplification and detection was done using the VECTASTAIN Elite ABC Kit (Burlingame, CA, USA) and the Metal Enhanced DAB Substrate Kit. Counterstaining was done using Harris’ modified hematoxylin.

### 4.13. Assay of SIRT1 Activity

The BIOMOL (Enzo Life Sciences, Farmingdale, NY, USA) assay was performed as previously described [74]. Recombinant SIRT1 (0.5 U) was used for each reaction, and the β-NAD and FdL-p53 peptide concentrations used were 200 μM and 20 μM, respectively. The vehicle or test compound (0.5 µL) was added to each 50 μL reaction mixture. Fluorescence values corresponding to SIRT1 activity were calculated by subtracting parallel reactions in the absence of β-NAD from those in the presence of β-NAD (F_corrected_ = F_+NAD_ − F_-NAD_). The PNC1-OPT assay was performed as done previously [74,75,76]. Each reaction was performed using ~2.5 μg of recombinant SIRT1 and ~1.25 μg of yPNC1, respectively. The sequence of native peptide substrate used in this assay, corresponding to H3K9ac, was “Ac-TARK(ac)STG-NH2”. Concentrations of β-NAD and H3K9ac peptide were 100 μM and 20 μM, respectively. Fluorescence values corresponding to SIRT1 activity were calculated by subtracting parallel reactions in the absence of β-NAD from those in the presence of β-NAD (F_corrected_ = F_+NAD_ − F_-NAD_).

### 4.14. Statistical Analysis

All experiments were carried out at least three times and statistical analysis was carried out using Student’s T test analysis. Furthermore, for all applicable datasets, one-way ANOVA analysis was also carried out to reveal significance with *p*-value < 0.0001 unless otherwise stated.

## Figures and Tables

**Figure 1 molecules-25-03849-f001:**
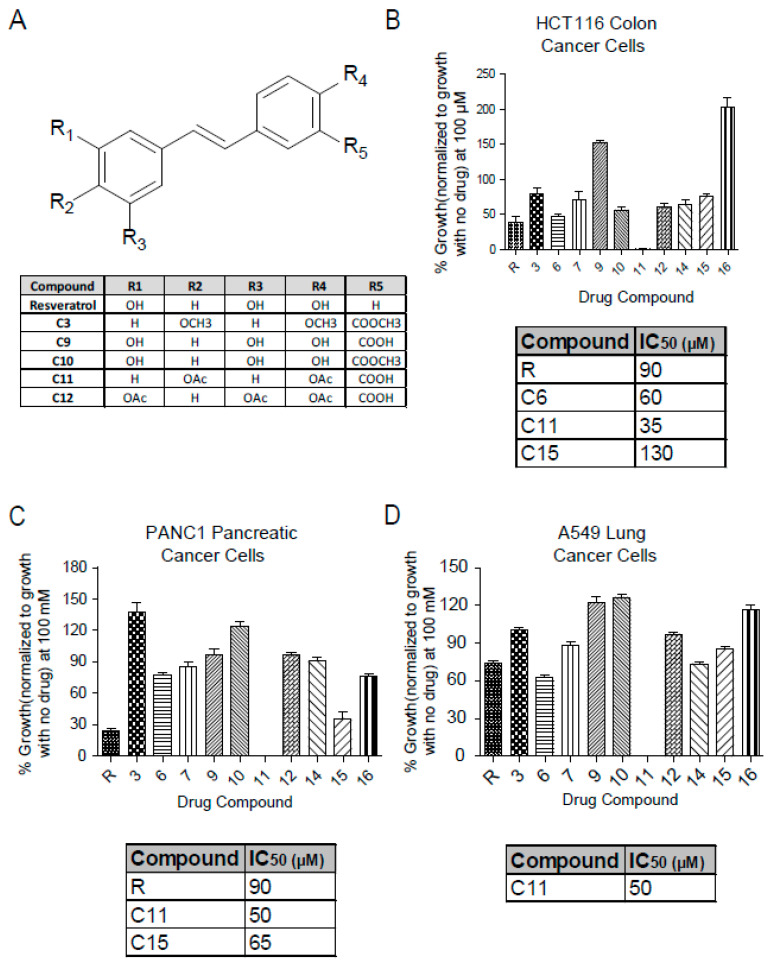
Chemical structure of resveratrol and derivatives and their effect on cell viability. (**A**) Table showing the chemical structure of parent compound resveratrol and some of the derivatives obtained from Dr. Carlos Velazquez Martinez from the Pharmacy department at the University of Alberta (For full chemical structures and preparation please see Reference [24]) (**B**–**D**) Analysis of cell viability with resveratrol and derivatives at 100 µM in colon cancer (HCT-116), pancreatic cancer (PANC1), and lung cancer (A549) cell lines. For IC_50_ calculation, cells were treated with concentrations between 0 to 200 µM (only effective compounds summarized in a table and can be seen in Appendix A). *N* = 8–16 with *p* values from < 0.05 to 0.001 for comparisons of compounds with resveratrol. If not stated, *p*-value > 0.05. One-way ANOVA analysis of (**B**–**D**) revealed significance with *p* values < 0.0001.

**Figure 2 molecules-25-03849-f002:**
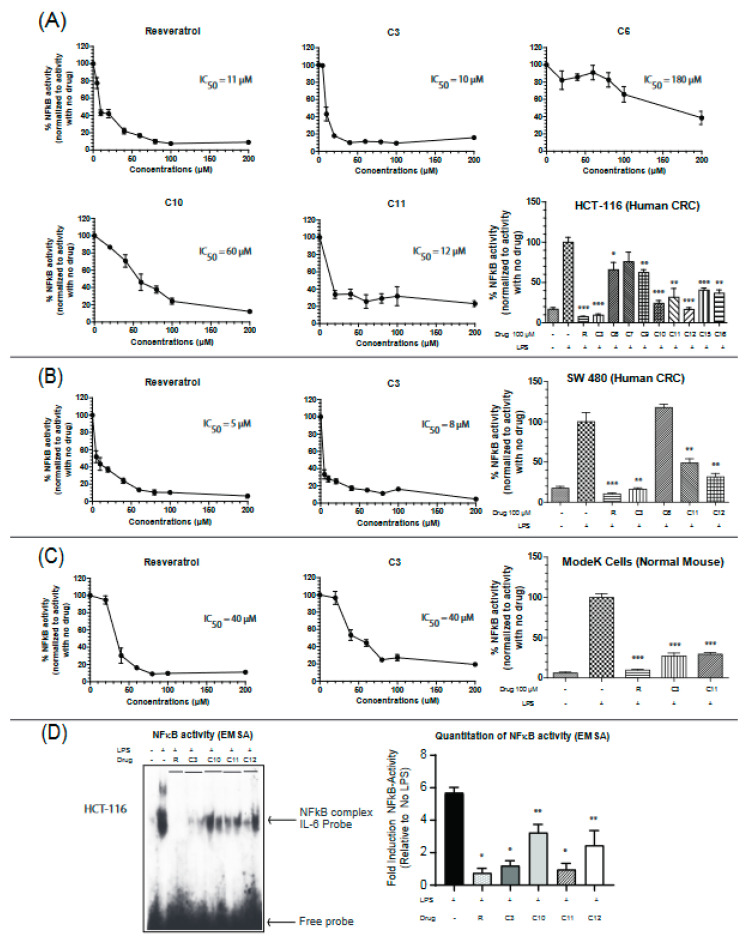
Resveratrol and derivatives inhibit NFκB activity in colon cancer and normal cell lines. NFκB activity was determined in (**A**) HCT-116, (**B**) SW 480, and (**C**) ModeK cells by a dual-luciferase assay with firefly luciferase under the control of NFκB target sequence on IL-6 promoter and Renilla-Luciferase used as an internal control. Cells were treated with drugs for 24 h before being stimulated with lipopolysaccharide (LPS) (1.5 mg/mL) for 4–6 h to ensure NFκB activation. Normalization was done against LPS-stimulated cells (100% activity). NFκB activity was determined by the ratio of Firefly-Luc/Renilla-Luc. The concentrations used for IC_50_ calculations ranged from 5 µM to 200 µM. The bar graphs show a comparison of the effectiveness of the different drugs at 100 µM. *n* = 3–8. (**D**) NFκB activity assessed by binding to the IL-6 promoter sequence DNA using the electrophoretic mobility shift assay (EMSA) technique. HCT-116 cells were grown to 60% confluency, treated with drugs for 24 h, and then stimulated with LPS (1.5 mg/mL) for 4 h in serum-free media. Two μg nuclear extracts were incubated with duplex DNA specific to NFκB (in duplicate). *N* = 4–6 * *p*-value < 0.0001, ** *p*-value < 0.003. If not stated, *p*-value > 0.05. For (**D**), *p*-value comparisons were made with respect to LPS-treated cells.

**Figure 3 molecules-25-03849-f003:**
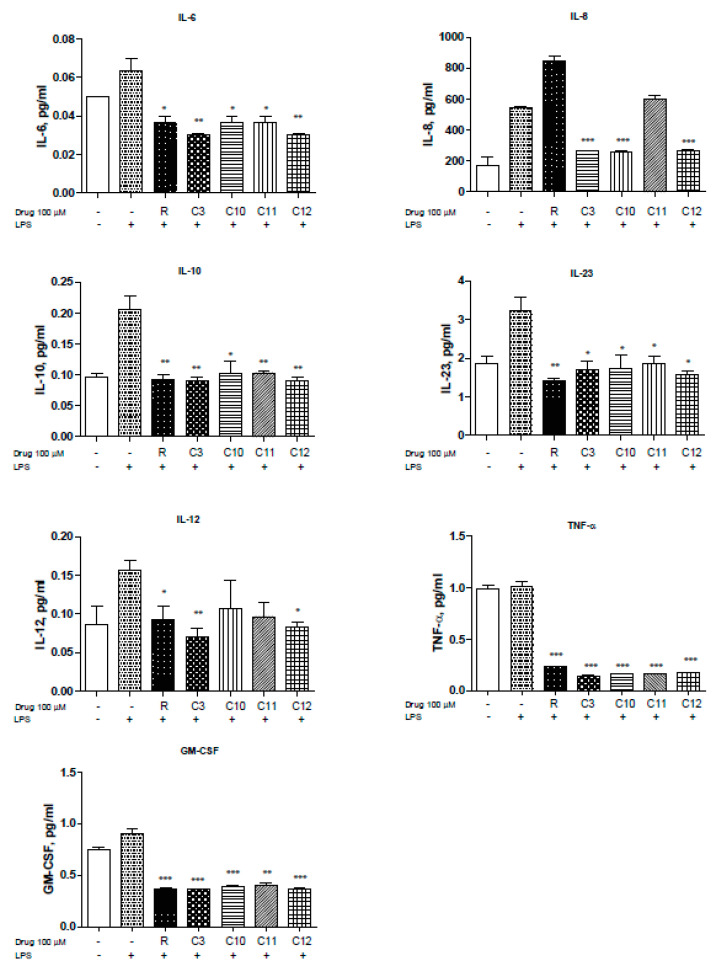
Resveratrol and derivatives reduce cytokine production in human colon cancer cell line HCT-116. Cytokine and chemokine levels were differentially reduced with resveratrol and derivatives. Briefly, HCT-116 cells were allowed to grow in 6-well plates to 60% confluency, treated with drugs (100 µM) for 24 h, and then stimulated with LPS (1.5 mg/mL) for 4 h. The supernatant (500 µL) was collected and sent to Eve technologies (https://www.evetechnologies.com/technology.php). *N* = 3. * *p*-value < 0.05, ** *p*-value < 0.01, and *** *p*-value < 0.001. If not stated, *p*-value > 0.05.

**Figure 4 molecules-25-03849-f004:**
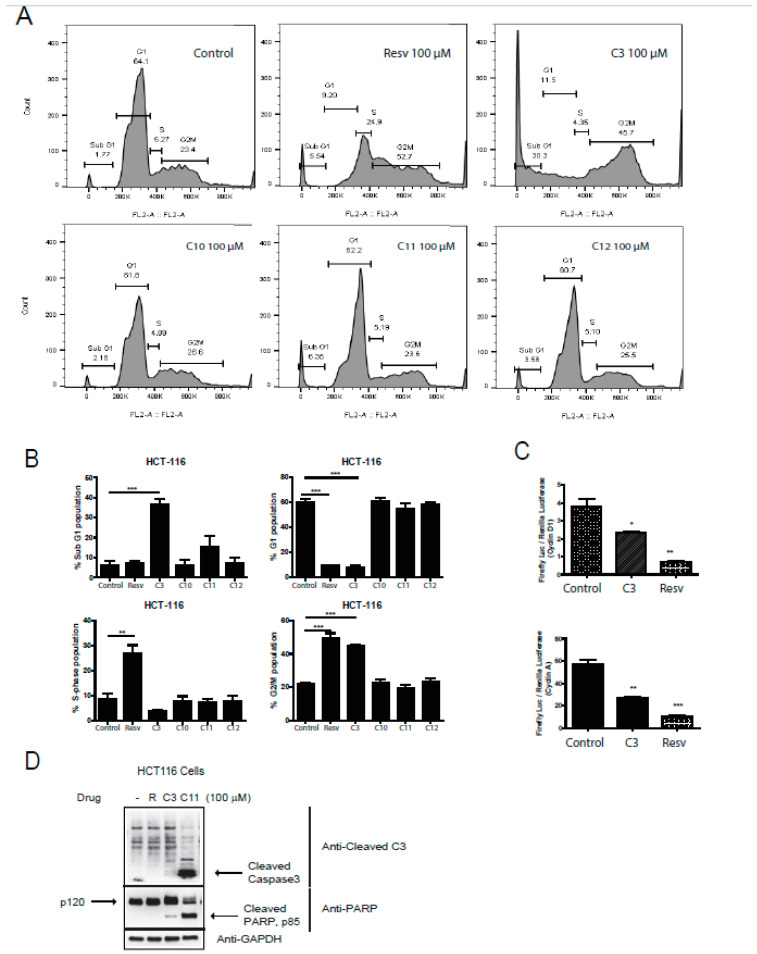
Effect of resveratrol and derivatives on cell cycle progression in human colon cancer and normal cell lines. (**A**) Cell cycle analysis in HCT-116 cells using propidium iodide (PI) staining followed by fluorescent activated cell sorting (FACS) analysis. In HCT-116 colon cancer cell line, resveratrol (100 µM) appears to induce S-phase and G2/M arrest whereas C3 (100 µM) significantly induces cell death and a G2/M arrest. *N* = 3–6. (**B**) Lower panels are quantitation of above cell cycle profiles. (**C**) Induction of cyclin expression was determined in HCT-116 cells by a dual-luciferase assay with firefly luciferase (Luc) under the control of a promoter to drive expression of cyclin D1 and cyclin A, and renilla-ruciferase used as an internal control. Cells were treated with drugs for 24 h. Normalization was done against non-treated cells (100% activity). The induction of cyclins was determined by the ratio of firefly Luciferase/Renilla-Luc. Experiments were done in triplicates. * *p*-value < 0.05, ** *p*-value < 0.01, and *** *p*-value < 0.001. If not stated, *p*-value > 0.05. (**D**) HCT116 cells were treated with the indicated drugs at 100 µM and immunoblotted for cleaved caspase 3 and PARP, markers for cell death. GAPDH was utilized as a loading control.

**Figure 5 molecules-25-03849-f005:**
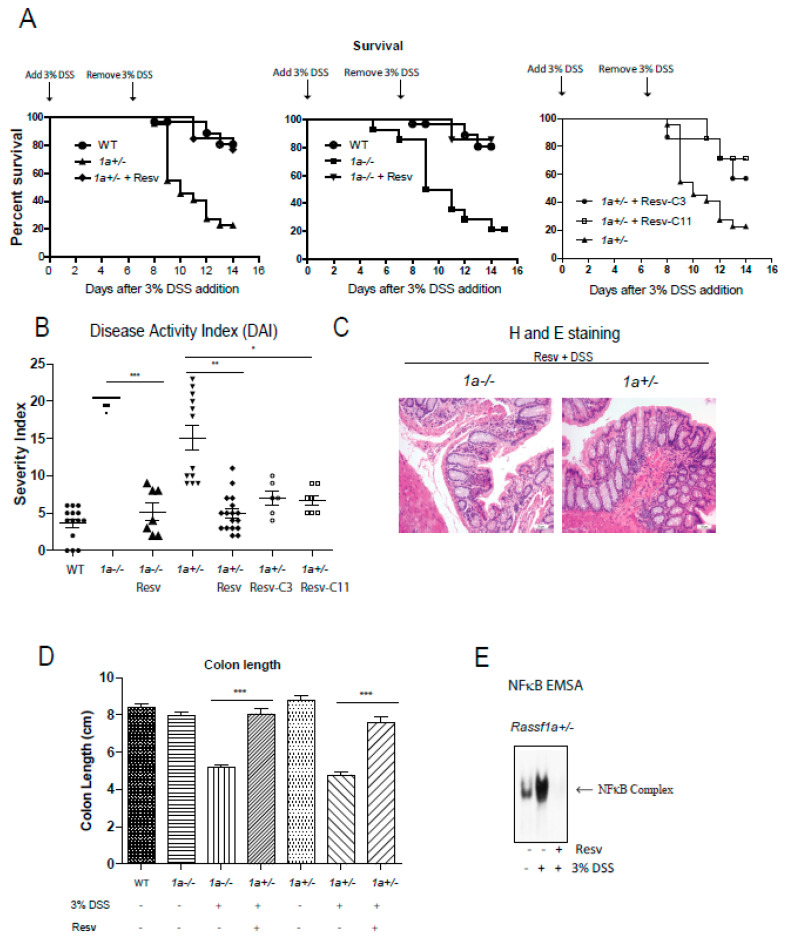
*Rassf1a^−/−^* and *Rassf1a^+/−^* knockout mice on resveratrol show decreased susceptibility to disease and enhanced survival. (**A**) Male mice were pre-fed with a resveratrol diet (containing 2 mg resveratrol/g of food) for two weeks prior to DSS. Three percent DSS addition was marked to day 0 and left for 7 days then mice were given regular drinking water for another 7 days. Tissue harvesting was carried out on day 8.5. Shown here is a Kaplan-Meier curve monitoring % survival following DSS treatment. For both the *1a^−/−^* and *Rassf1a^+/−^* knockout mice on resveratrol and DSS, *p*-value < 0.0001 when compared to counterpart on regular diet and DSS. For *Rassf1a^+/−^* mice injected with Resv-C3 *p*-value is 0.01 and *p*-value is 0.005 for Resv-C11 injected mice when compared to counterpart *Rassf1a^+/−^* mice on regular diet and DSS. Resveratrol fed *Rassf1a^−/−^* and *Rassf1a^+/−^* knockout mice showed survival rates comparable to wild type when given DSS in drinking water. *n* > 7. (**B**) Resveratrol fed mice (as in (**A**)) or Resv-C3 and Resv-C11 injected *Rassf1a^−/−^* and *Rassf1a^+/−^* knockout mice (intraperitoneally at 0.3 mmol/kg body weight every two days) showed less susceptibility to disease when given DSS in drinking water. Disease activity indices (DAI) accounted for several parameters including piloerection, bloated gut, movement, rectal bleeding, hunching, diarrhea, and weight loss. A numerical value of 1 to 5 was given with 5 being severe. If an animal was found dead 5 points added to previous day DAI. *N* > 7 * *p*-value < 0.05, ** *p*-value < 0.01, and *** *p*-value < 0.001. If not stated, *p*-value > 0.05. (**C**) Representative figures of the descending colon (longitudinal cross-section) stained with hematoxylin and eosin (H&E) for knockout mice fed with resveratrol reveal a well-defined crypt structure. (**D**) Colon length was measured at day 8.5 of DSS treatment. Resveratrol maintained “normal” (wild type) lengths indicative of non-injured healthy colon. *N* > 7 * *p*-value < 0.05, ** *p*-value < 0.01, and *** *p*-value < 0.001. If not stated, *p*-value > 0.05. (E) NFκB electrophoretic mobility shift assay (EMSA) on bone marrow-derived macrophage (BMDM) nuclear extracts as described earlier reveals resveratrol inhibition of NFκB activity in *Rassf1a^+/−^* mice.

**Figure 6 molecules-25-03849-f006:**
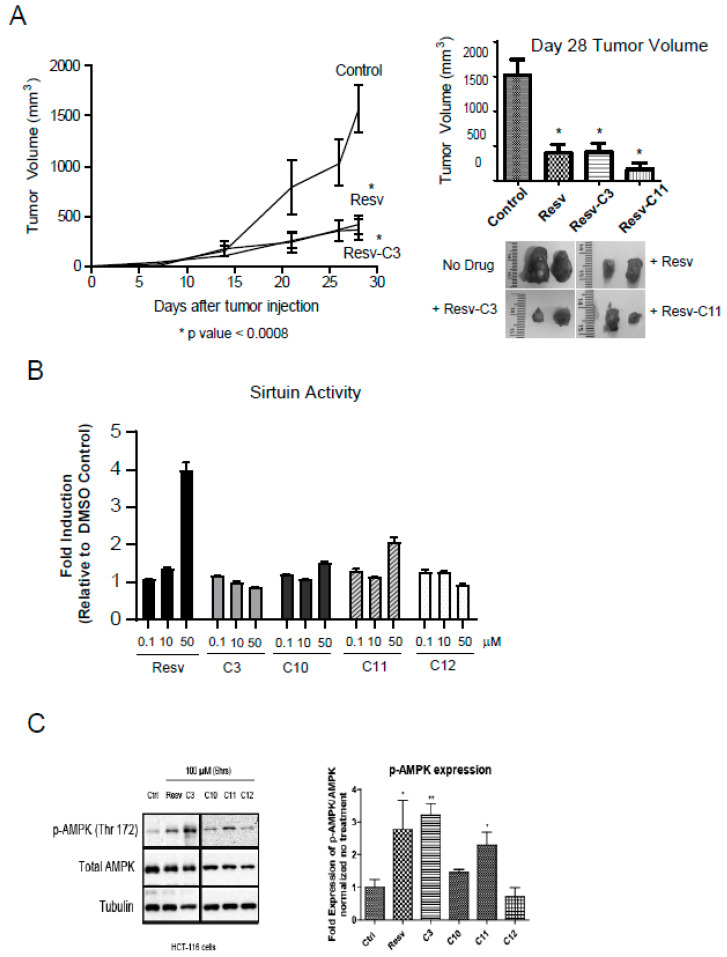
Resveratrol-treated mice reveal reduced tumor burden in a xenograft model. (**A**) HCT116 colon cancer cells were injected subcutaneously into the flanks of athymic mice and tumor formation was monitored over 28 days. Resveratrol fed mice (as in (**A**)) at 4 mg resveratrol/g (or 0.017 mmols/g) of food or Resv-C3/Resv-C11 injected intraperitoneally at 0.3 mmol/kg (or 68.4 mg/kg) body weight every two days we examined for tumor formation. Left panel, the time course of tumor formation (note that Resv-C11 graph overlapped with resveratrol diet and Resv-C3 and was excluded from the plot for clarity). Right panel, day 28 tumor volumes with representative pictures of tumors excised from the animal. For * *p* values range from 0.0001 to 0.0005 and *n* = 10–15. (**B**) Analysis of sirtuin assay using various aspirin-resveratrol derivatives. *n* = 3–4 replicates. (**C**) HCT116 cells with the drugs for 24–48 h and then harvested and Western blot for pT172 AMPK and total AMPK as shown in the left panel and quantitated in the right panel. Analysis was performed using an Image J open source program.

**Table 1 molecules-25-03849-t001:** Summary of biological properties of resveratrol and derivatives. Table showing a summary of the biological properties characterized in this study for the most effective salicylate resveratrol derivatives as determined in this study and other publications.

Compound	Cell Viability ^†^ IC50	NFκB Inhibition ^†^ IC50	Inhibits Tumor Formation	DNMT1 * IC50	DNMT3B * IC50	Sirtuin ^#^ Activation (Fold Change at 50 µM)	AMPK ^#^ Activation (Fold Change at 100 µM)
Resveratrol	90 μM	11 μM	YES	> 300 μM	65 μM	3.97	2.7
Resv-C3	>200 μM	10 μM	YES	N.I.	> 300 μM	0.82	3.2
Resv-C9	>100 μM	> 200 μM	No	N.I.	52 μM	0.85	N.D.
Resv-C10	>100 μM	60 μM	YES		62 μM	1.53	~1.5
Resv-C11	35 μM	12 μM	YES	N.I.	190 μM	1.94	2.4
Resv-C12	>100 μM	20 μM	N.D.		215 μM	0.92	~0.8

**^†^** As determined in HCT-116 cells; * Reported in previous publications; ^#^ Fold change is depicted in relation to non-resveratrol treated cells; N.I.: No inhibition at maximum tested concentration > 300 µM. N.D.: not determined.

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
