# Peer review of "Resveratrol and Resveratrol-Aspirin Hybrid Compounds as Potent Intestinal Anti-Inflammatory and Anti-Tumor Drugs"

_molecules, 2020, doi:10.3390/molecules25173849_

Round 1

Reviewer 1 Report

The manuscript entitled "Resveratrol and resveratrol-aspirin hybrid compounds as potent intestinal anti-inflammatory and anti-tumor drugs" by M. Salla et al. describes the biological activity of resveratrol-hybrid compounds. Many health-promoting benefits have been connected with resveratrol in the treatment of cardiovascular diseases, cancer, diabetes, inflammation, neurodegeneration, and diseases connected with aging. However, resveratrol has a limited bioavailability, due to a rapid metabolism in the intestine and liver. Therefore, the effects of resveratrol in humans are often tiny and not always reproducible. Resveratrol can be used as a scaffold compound to design and synthesize resveratrol analogs. The aim would be to create resveratrol analogs that have an increased bioavailability, potency and efficacy in comparison with resveratrol so that lower concentrations could be used as therapeutics in vivo. The authors used this strategy and synthesized and testet various resveratrol-aspirin hybrid compounds. There results show that the reseveratrol-aspirin derivatives inhibit NF-kB activation, cytokine production and the groth rate of cancer cells in vivo. The manuscript describes an interesting peace of work. However, a number of functional and technical questions need to be addressed.

  1. Resveratrol is toxic to many cell types at concentrations higher than 20 microM. The authors used higher concentrations. I assume that the description “growth … at 100 mM” (Fig. 1C and D) is a typo and 100 microM were used. Nevertheless, a resveratrol concentration of 100 microM kills many cells.

  1. For their experiments concerning the effect on resveratrol-aspirin hybrid derivatives on NF-kB activity, the authors stimulated the cells with LPS to activate NF-kB and then tested whether the resveratrol compounds are able to reduce the LPS-initiated NF-kB activation. Thus, the impact of resveratrol on an already activated NF-kB was investigated. Recently, it has been shown that resveratrol activates NF-kB and stimulates expression of the NF-kB target gene encoding IL-8 (Thiel et al., Pharmacol Res. 2018). Thus, it would be of interest to analyze whether the resveratrol-aspirin hybrid derivatives activate NF-kB in the absence of LPS

  1. The MTT assay measures the reduction of MTT to formazan by NAD(P)H-dependent oxidoreductase enzymes and can be used as an indirect measure for growth or apoptosis. The results should therefore describe their results as “MTT reduction”.

  1. In vitro DNA-protein binding assays (EMSA) are an outdated method. The authors should either delete these results or perform chromatin immunoprecipitation experiments.

  1. The Rassf1a(-/-) mouse model needs to be described in more detail.

Author Response

Answers to Reviewer #1

We thank reviewer #1 for recognizing the relevance of our work and the support our data provides for the conclusions. We have carefully gone over the research design and methods as well as the results to improve the reading of those sections. We have addressed his/her comments below:

  1. Resveratrol is toxic to many cell types at concentrations higher than 20 microM. The authors used higher concentrations. I assume that the description “growth … at 100 mM” (Fig. 1C and D) is a typo and 100 microM were used. Nevertheless, a resveratrol concentration of 100 microM kills many cells.

SBaksh: The reviewer has made relevant comments. We do apologize for the mistake of labelling the concentration as “100 mM.” Indeed, it should be labelled as “100 M” and this has been corrected. For the toxicity issue, we have addressed this in Figure 4 and show that only C3 can induced a sub-G1 increase to reflect a pro-apoptotic phenotype. C11 does severely affect the growth of HCT116 and other cell types mainly by arresting in G1 as shown in Figure 4. This is discussed in section 2.4 of the manuscript text.

  1. For their experiments concerning the effect on resveratrol-aspirin hybrid derivatives on NF-kB activity, the authors stimulated the cells with LPS to activate NF-kB and then tested whether the resveratrol compounds are able to reduce the LPS-initiated NF-kB activation. Thus, the impact of resveratrol on an already activated NF-kB was investigated. Recently, it has been shown that resveratrol activates NF-kB and stimulates expression of the NF-kB target gene encoding IL-8 (Thiel et al., Pharmacol Res. 2018). Thus, it would be of interest to analyze whether the resveratrol-aspirin hybrid derivatives activate NF-kB in the absence of LPS.

SBaksh: From the data in Figure 2, resveratrol or its derivatives do not activate NFkB but inhibit NFkB. Contrary to what is observed in the Pharmacol Res. publication in 2018, we do not and have never observed activation of NFkB by resveratrol or its derivatives.

3. The MTT assay measures the reduction of MTT to formazan by NAD(P)H-dependent oxidoreductase enzymes and can be used as an indirect measure for growth or apoptosis. The results should therefore describe their results as “MTT reduction”.

SBaksh: The reviewer is correct and we have added a sentence to reflect that at the bottom of page 4 in section 2.1

4. In vitro DNA-protein binding assays (EMSA) are an outdated method. The authors should either delete these results or perform chromatin immunoprecipitation experiments.

SBaksh: We respectfully disagree with the reviewer that EMSA is an outdated methodology. On the contrary, it is a robust measure of the ability of nuclear fraction proteins to associate with DNA. It measures direct binding to DNA in an in vitro system. Chromatin immunoprecipitation experiments are more in vivo but also have many limitations and need specialized antibodies. We have improved the contrast of Figure 2D with new quantitation to include a second run as seen in Supplementary Figure S4.

5. The Rassf1a(-/-) mouse model needs to be described in more detail.

SBaksh: All the required information for this model is in reference #36 and have added the sentence “We provided evidence to support the role of RASSF1A as a negative regulator of inflammation when, if lost, will result in uncontrolled inflammation in the colon.”

Reviewer 2 Report

The present manuscript analyses the potential beneficial effect of resveratrol and resveratrol-aspirin hybrid compounds on inflammation and cancer in vivo and in vitro. This is an interesting topic which has been scarcely studied; however, a major concern arises from the fact that non-realistic doses have been used and that a similar effect was detected on normal and cancer cells on cell proliferation. Moreover, a deeper study on certain aspects to ensure the effect defined must be performed (see below). Other aspects should also be checked:

  • Results: Section 2.1: Please, note that the corresponding bar for C11 is missing in figures 1C, 1D, and S2A.

If compounds are toxic in non-cancer cells, where is the advantage of using these new synthetic compounds? An explanation about the relevance of the effective doses used would be appreciated by the reader. Could ever colon, breast, pancreas, etc cells be exposed to those doses? Please, explain.

Section 2.2: why were just colon cells used? Why were HCT116 and SW480 cells selected? Please, explain.

How do the authors explain than C11, which was the best inducing cell damage, showed worst results than C3?

Why was not C12 tested at different concentrations as showed better results than C6 and it is later used (section 2.3.)? Please, consider analysing C12 effect on NFkB at different concentrations.

Which was the rationale to select 100 µM for the compounds?

Which was the rationale to select the LPS dose? Please, support with a reference and explain.

Statistical differences seem to be missed in Figure S4 for RANTES, MCP3, PDGF, IL6, IL8, and IP10. Please, check.

The EMSA image in figure 2D must be improved.

Section 2.4: In this section cell cycle arrest is assayed. I would suggest to rename this section as section 2.2. as it is more connected to MTT analysis. In addition, the authors must consider that to demonstrate apoptosis induction caspases (at least 3) and Bcl-2 anti- and pro-apoptotic proteins (Bcl-2, Bcl-x, Bax, etc. must be assayed.

Section 2.5: Basic parameters related to the animals are missing (initial and final body weight, food intake, colon weight, etc.)

In figure 5S, what does it mean an increase in PCNA values in WT and KO mice treated with resveratrol? How was apoptosis induction in the mentioned animals (since colon is an epithelial tissue the balance between proliferation and death is very important)?

How was the aspect of colon mucosa in animals treated with C3 and C11? Similarly, how were colon length and NFkB activity in the mentioned animals?

Section 2.6: Which was the effect on tumour volume of C10 and C12 which are also active in modulating AMPK? Why were not C10 and C12 assayed? Please, explain.

Table 1 why was C9 included since this compound has not been assayed in detail in the present manuscript? What about C10 or C12?

  • Discussion: Were doses used in animals and cells comparable?

This section could be enriched by comparing the present result s with similar studies. Why was 100 µM the concentration selected for the compounds to be tested.

  • Material and methods: Section 3.2: Please, provide the number of cells seeded for each assay.

Section 4.6: Please, briefly explain how nuclear extracts were prepared.

Section 4.7: how were animals sacrificed?

Western blot procedure to analyse AMPK is missing

Statistical section is missing. Please, consider to perform ANOVA as all comparisons among all groups could be showed and the efficacy of compounds compared among them. In this line, differences among compounds could be highlighted and discussed accordingly.

Author Response

Answers to Reviewer #2

We thank reviewer #2 for recognizing the relevance of our work. We have carefully gone over the research design and results as well as the results to improve the reading of those sections. We have addressed his/her comments below: 

  • Results: Section 2.1: Please, note that the corresponding bar for C11 is missing in figures 1C, 1D, and S2A.

SBaksh: The results for C11 are not missing but C11 dramatically reduces growth of many cell types. We generally observe only 3-5% growth when compared to no drug treated cells upon C11 incubation. Thus, C11 treatment results in > 95% reduction in growth (mainly resulting in G1 arrest with little apoptotic induction. We have added the statement “C11 treatment results in > 95% inhibition of growth by promoting a robust G1 arrest to inhibit growth” to section 2.1, top of page 5.

If compounds are toxic in non-cancer cells, where is the advantage of using these new synthetic compounds? An explanation about the relevance of the effective doses used would be appreciated by the reader. Could ever colon, breast, pancreas, etc cells be exposed to those doses? Please, explain.

SBaksh: The advantage of presenting data on these synthetic new compounds are their multiple effects on growth, NFkB, sirtuin and AMPK activities. The multiple readouts presented here have not been done systematically in previous reports and our results will be very useful to others entering into this field. The activities towards these targets are encouraging to help in designing the next generation derivaties.

Section 2.2: why were just colon cells used? Why were HCT116 and SW480 cells selected? Please, explain.

SBaksh: We have added the sentence “Colon cancer cells are very responsive to LPS treatment and display robust NFkB activities.” To explain why we selected these two cell types for our assays.

How do the authors explain than C11, which was the best inducing cell damage, showed worst results than C3?

SBaksh: C3 induced the strongest cell death response and NFkB inhibition and C10/C11/C12 induced the strongest G1 arrest. What I can observe from our results is that C3 and C11 had similar in vivo effects (tumor assay and DSS-induced inflammation inhibition) while most of their in vitro inhibition were quite similar (except for growth inhibition). So, I am not sure what the reviewer meant by C11 “showed worst results than C3”?

Why was not C12 tested at different concentrations as showed better results than C6 and it is later used (section 2.3.)? Please, consider analysing C12 effect on NFkB at different concentrations.

SBaksh: C12 was tested at different concentrations as seen below. These two curves has been added to Supplementary Figure S4A.

Which was the rationale to select 100 µM for the compounds?

SBaksh: That concentration was utilized based on the IC50 curves in the various figures.

Which was the rationale to select the LPS dose? Please, support with a reference and explain.

SBaksh: LPS dose was based on our previous work in reference 36 and based on literature documentation of LPS use.

Statistical differences seem to be missed in Figure S4 for RANTES, MCP3, PDGF, IL6, IL8, and IP10. Please, check.

SBaksh: We have clearly indicated p values in the figure and stated “If not stated, p-value > 0.05.”

The EMSA image in figure 2D must be improved.

SBaksh: We have improved the contrast of Figure 2D with new quantitation to include a second run as seen in Supplementary Figure S4.

Section 2.4: In this section cell cycle arrest is assayed. I would suggest to rename this section as section 2.2. as it is more connected to MTT analysis. In addition, the authors must consider that to demonstrate apoptosis induction caspases (at least 3) and Bcl-2 anti- and pro-apoptotic proteins (Bcl-2, Bcl-x, Bax, etc. must be assayed.

SBaksh: We have corrected the title of section 2.4 to “4 Resveratrol and derivatives arrested HCT-116 colorectal cancer cell line at different stages of the cell cycle.”

Section 2.5: Basic parameters related to the animals are missing (initial and final body weight, food intake, colon weight, etc.)

SBaksh: We have added “(males at 23-25 g) to the mouse experiment section of methods. We have tracked food consumption extensively as well as final body weights. The sentence,     Generally, if the animal has 25 g to start, at the end of the 14 day treatment with severe inflammation, the body weight is generally around 14-16 g. We did not include this data as it was too extensive to include. 

In figure 5S, what does it mean an increase in PCNA values in WT and KO mice treated with resveratrol? How was apoptosis induction in the mentioned animals (since colon is an epithelial tissue the balance between proliferation and death is very important)?

SBaksh: We have utilized PCNA staining to monitor the proliferation of epithelial cells and hence a marker of active proliferation and healthiness of the colonic tissue. We have extensively utilized this in our publication in reference #36. Inflammation injury affects the growth of the epithelial layer and decreased epithelial restitution and would healing. Apoptotic injury does occur but it is mainly a lack of epithelial restitution that leads to injury during DSS treatment.

How was the aspect of colon mucosa in animals treated with C3 and C11? Similarly, how were colon length and NFkB activity in the mentioned animals?

SBaksh: Because of how extensive our data was, we did not present data with C3 and C11. With both these derivative, inflammation injury was dramatically down resulting in increased colon length and reduced NFkB activity.

Section 2.6: Which was the effect on tumour volume of C10 and C12 which are also active in modulating AMPK? Why were not C10 and C12 assayed? Please, explain.

SBaksh: Both C12 and C10 had modest effects on activating AMPK as shown in Figure 6C. They did not have an effect on tumor inhibition as carried out in Figure 6A and the data was not presented. I have now added “We did not observe any protective effect for Resv-C10 or Resv-C12.” Section 2.6 was mainly focused on C3 and C11 as they showed the most effect on growth inhibition.

Table 1 why was C9 included since this compound has not been assayed in detail in the present manuscript? What about C10 or C12?

SBaksh: We apologize for not present all the data. We have not edited Table 1 and have added data to Supplementary Figure S4 with NFkB data for C10 and C12 and created Supplementary Figure S6 with C10 tumor data. C12 was not pursued in the tumor data due to no effect on growth in multiple cell types.

  • Discussion: Were doses used in animals and cells comparable?

SBaksh: Yes it was as we tried to match that.

This section could be enriched by comparing the present results with similar studies. Why was 100 µM the concentration selected for the compounds to be tested.

SBaksh: We have added comparison with other animal experiments on pages 10 and 11 and some comparisons with cell based assays.

  • Material and methods: Section 3.2: Please, provide the number of cells seeded for each assay.

SBaksh: These assays were carried out in 6 well dishes whereby cells were cell seeding is indicated for each assay as appropriate.

Section 4.6: Please, briefly explain how nuclear extracts were prepared.

SB: All extracts were prepared using the NEPER extraction kit from ThermoFisher Scientific. This is a standard kit that utilizes proprietory reagents. My guess is that the first step is cell swelling in hypotonic buffer to burst the plasma membrane thus releasing the cytoplasmic components. This is followed by a high salt extraction to burst the nuclear membrane releasing the nuclear fraction. This is based on previous home made recipes I have used in the past.

Section 4.7: how were animals sacrificed?

SB: All animals were sacrificed by CO2 inhalation as outlined in our protocol. This has been added to the methods section.

Western blot procedure to analyse AMPK is missing

SB: We have added the sentence “Analysis was preformed using a an Image J open source program.” Thanks to the reviewer for pointing this out to us.

Statistical section is missing. Please, consider to perform ANOVA as all comparisons among all groups could be showed and the efficacy of compounds compared among them. In this line, differences among compounds could be highlighted and discussed accordingly.

SBaksh: We have added the sentence:” Statistical Analysis. All experiments were carried out at least three times and statistical analysis was carried out using Student’s T test analysis.” to section 4. We have also added the phrase. “One-way ANOVA analysis of was also carried out on all relevant data set to reveal significance with p value < 0.0001.

Reviewer 3 Report

The original article by Salla M et al set out to evaluate resveratrol derivatives for their ability to inhibit growth of cancer cells. The paper is presented well with data easily interpretable. Sufficient mechanistic studies were performed with in vivo data strongly supporting their claims about C3 and C11 compounds. This study will be of broad interest to the readership of Molecules and cancer research community in general. I only have a few minor comments:

1. Can the authors speculate why C3 has such as strong in vivo anti-tumor effects yet shows no such anti-proliferative effect in vitro (Fig. 1)?

2. I may be off, but in Figure 2D, it does not appear that there is formation of NFkB complex under R treatment. Could this be clarified?

Author Response

Answers to Reviewer #3

We thank reviewer #3 for recognizing the relevance of our work and the conclusions drawn. We have carefully gone over the various sections to improve the reading of those sections. We have addressed his/her comments below:

  1. Can the authors speculate why C3 has such as strong in vivo anti-tumor effects yet shows no such anti-proliferative effect in vitro (Fig. 1)?

SBaksh: We do not know at this time but it is an interesting observation. Certainly the effect on NFkB and AMPK could be enough to sway the proliferative pathways during tumorigenesis to an anti-proliferative state. In cell culture, you do not see the effect of modulation of the tumor microenvironment as you would see it in an animal model.

  1. I may be off, but in Figure 2D, it does not appear that there is formation of NFkB complex under R treatment. Could this be clarified?

SBaksh: We have improved the contrast of Figure 2D with new quantitation to include a second run as seen in Supplementary Figure S4. Lanes 3 and 4 from the left come from resveratrol treated cells so I think the reviewer may have misread the lanes. We hope we have not alleviated the confusion associated with this panel.

Round 2

Reviewer 1 Report

The authors have addressed the concerns of the reviewer. Therefore, I recommend to publish the manuscript.

Author Response

Answers to Reviewer #1

We thank reviewer #1 for recognizing the relevance of our work and the support our data and accepting our responses.

Reviewer 2 Report

The revised manuscript has been improved, but some questions have not been answered; some (major) aspects that remain:

  • Results: Section 2.1: An explanation about the relevance of the effective doses used would be appreciated by the reader. Could ever colon, breast, pancreas, etc cells be exposed to those doses? Please, explain.

Section 2.2: Which was the rationale to select 100 µM for the compounds? Are these doses achievable in humans?

Section 2.3: This reviewer did not get the revised figures in the revised manuscript (just supplementary figures in a separated file); therefore, I do not know how the new image in figure 2D looks like, but this original image (EMSA image in figure 2D) must be improved.

Section 2.4: Please, note that the section has not been renamed.

The authors must consider that to demonstrate apoptosis induction caspases (at least 3) and Bcl-2 anti- and pro-apoptotic proteins (Bcl-2, Bcl-x, Bax, etc. must be assayed.

Section 2.5: Please, include the final body weight and food intake.

In figure 5S, what does it mean an increase in PCNA values in WT and KO mice treated with resveratrol? How was apoptosis induction in the mentioned animals (since colon is an epithelial tissue the balance between proliferation and death is very important)?

How was the aspect of colon mucosa in animals treated with C3 and C11? Similarly, how were colon length and NFkB activity in the mentioned animals?

  • Material and methods: Western blot procedure to analyse AMPK is missing

Author Response

We thank reviewer #2 for recognizing the relevance of our work and helpful comments to improve our manuscript. We have attached a detailed summary in a PDF file.
